# The effect of a behaviour change intervention on the diets and physical activity levels of women attending Sure Start Children's Centres: results from a complex public health intervention

Janis Baird,[1] Megan Jarman,[1,2] Wendy Lawrence,[1,2] Christina Black,[1] Jenny Davies,[3] Tannaze Tinati,[1] Rufia Begum,[1,2] Andrew Mortimore,[3] Sian Robinson,[1] Barrie Margetts,[4] Cyrus Cooper,[1,2] Mary Barker,[1] Hazel Inskip[1]

For numbered affiliations see end of article.

**Correspondence to**
Dr Janis Baird;
jb@mrc.soton.ac.uk

## ABSTRACT

**Objectives:** The UK government's response to the obesity epidemic calls for action in communities to improve people's health behaviour. This study evaluated the effects of a community intervention on dietary quality and levels of physical activity of women from disadvantaged backgrounds.

**Design:** Non-randomised controlled evaluation of a complex public health intervention.

**Participants:** 527 women attending Sure Start Children's Centres (SSCC) in Southampton (intervention) and 495 women attending SSCCs in Gosport and Havant (control).

**Intervention:** Training SSCC staff in behaviour change skills that would empower women to change their health behaviours.

**Outcomes:** *Main outcomes* dietary quality and physical activity. *Intermediate outcomes* self-efficacy and sense of control.

**Results:** 1-year post-training, intervention staff used skills to support behaviour change significantly more than control staff. There were statistically significant reductions of 0.1 SD in the dietary quality of all women between baseline and follow-up and reductions in self-efficacy and sense of control. The decline in self-efficacy and control was significantly smaller in women in the intervention group than in women in the control group (adjusted differences in self-efficacy and control, respectively, 0.26 (95% CI 0.001 to 0.50) and 0.35 (0.05 to 0.65)). A lower decline in control was associated with higher levels of exposure in women in the intervention group. There was a statistically significant improvement in physical activity in the intervention group, with 22.9% of women reporting the highest level of physical activity compared with 12.4% at baseline, and a smaller improvement in the control group. The difference in change in physical activity level between the groups was not statistically significant (adjusted difference 1.02 (0.74 to 1.41)).

**Conclusions:** While the intervention did not improve women's diets and physical activity levels, it had a protective effect on intermediate factors—control and self-efficacy—suggesting that a more prolonged

### Strengths and limitations of this study

- This complex public health intervention was delivered within existing health and social care services. This approach to delivery in the real world meant that the intervention was sustainable and could be continued after the study was completed.
- Data collected during interviews with women meant that important confounding factors and effect modifiers could be considered in analyses.
- Randomisation at the level of Sure Start Children's Centres was not feasible because of the frequent movement of staff between centres which would have led to contamination of control centres.
- A cluster randomised controlled trial was considered but the need for many clusters, each being an entire local authority area, was considered unfeasible by an independent advisory group of experts.

exposure to the intervention might improve health behaviour. Further evaluation in a more controlled setting is justified.

## INTRODUCTION

In response to the current epidemic of obesity and chronic disease, the Foresight committee has recommended change at many levels—personal, family, environmental and national.[1] The government response, outlined in *Healthy Lives, Healthy People*, emphasises the need to empower individuals to make healthy choices, recognising that reduction in obesity will only be achieved if the dietary and physical activity behaviour of the population improves.[2] A recent review

of evidence has demonstrated that government policy has led to a reduction in unhealthy behaviours but that these reductions have mainly been among groups of higher socioeconomic status and educational attainment.[3] Mothers have a strong influence on the health behaviour of their families, particularly in relation to diet, since they have considerable influence over food-related decisions within the family.[4] The nutritional status and health of mothers also influence the growth and development of infants during pregnancy and in postnatal life. Growth and development at these stages of the life course will influence the risk of chronic conditions such as cardiovascular disease and obesity in adulthood.[5]

Women of childbearing age from disadvantaged backgrounds are more likely to have poor-quality diets and are less likely to take part in regular physical activity, both of which are detrimental to their own health and to the growth and development of their children. The Southampton Women's Survey (SWS)[6] has demonstrated that women who are disadvantaged by leaving school with few or no educational qualifications eat a less varied and balanced diet than women with higher levels of educational attainment.[7] These findings have been confirmed in other mother-offspring cohorts.[8] The SWS has also shown a link between the quality of mothers' diets and the diets of their children: of SWS women who became pregnant, those with the least healthy preconception diets were found to be less likely to follow guidance on optimal patterns of infant feeding.[9] The quality of infant diet is an important determinant of childhood outcomes: diets of poor quality have been associated with increased fat mass[10] and lower IQ in longitudinal studies.[11] [12] Data from the SWS have also shown that women living in disadvantaged areas of Southampton are less likely to take part in regular strenuous exercise that would be beneficial to their health.[13]

Studies exploring barriers to healthy eating among disadvantaged women have shown that women who had the poorest quality diets felt that they lacked control over the food choices they made for themselves and their families.[14] [15] Numerous studies have confirmed the relationship of a higher sense of control and self-efficacy with better dietary behaviour and higher levels of physical activity.[16] [17] Efforts to improve the health of women from disadvantaged backgrounds need to take account of their lifestyle choices and address the barriers to healthy patterns of behaviour.

Recent reviews of evidence have provided useful insights into the features of behaviour change interventions associated with effectiveness in low income groups and women of childbearing age: providing information on the risks and benefits of health behaviours combined with goal setting and continued support after the initial intervention was more likely to lead to behaviour change.[18] [19] Similarly, a review of interventions that were effective in improving health behaviours in populations considered at risk of developing diabetes found that interventions most likely to be effective were those that targeted both diet and physical activity. In addition, interventions most likely to be effective used established behaviour change techniques and focused on a 'self-regulatory' approach such as goal setting and self-monitoring. The review also found that frequent contact, with the professional delivering the intervention, and engagement with social support were associated with effectiveness.[17] Consistent with this evidence from population-based studies of behaviour change in women from disadvantaged backgrounds, low-income groups and at-risk populations, evidence from clinical populations suggests that empowering patients to take control of their conditions has benefits in improving disease self-management for conditions such as arthritis and asthma.[20] It is believed that this sort of empowerment approach works because it increases patients' self-efficacy.[21] [22]

We have applied the principles of an empowerment approach to an intervention which aims to improve the health behaviour of women from disadvantaged backgrounds, a group in which there is an established link between low sense of control and self-efficacy with poor-quality diet. In accordance with the MRC guidance on complex interventions,[23] we developed the Southampton Initiative for Health, an intervention which aimed to improve the diets and physical activity levels of women from disadvantaged backgrounds. The intervention, which has been described in detail elsewhere, achieved its aims by training Sure Start Children's Centre (SSCC) staff, who work with women and children from disadvantaged families, in behaviour change techniques.[24] The logic model for the intervention demonstrates how improvements in sense of control and self-efficacy are intermediate outcomes in the path between the women's contact with trained SSCC staff and improvements in their diets and physical activity levels. Within the model, sense of control is defined as an individual's perception that they have control over their lives and self-efficacy is defined as an individual's belief that they are capable of carrying out a specific behaviour.

This paper describes the findings of an exploratory study evaluating the effect of the intervention on the diets and physical activity levels of women attending SSCCs in Southampton 1 year after the training had been delivered to SSCC staff within the city.

## METHODS

We carried out a before and after non-randomised controlled evaluation of the complex public health intervention. Randomisation at the SSCC level was not possible because staff and women were known to move frequently between centres. The intervention was introduced in the 14 SSCCs within Southampton and training of staff took place from May 2009 to May 2010. Implementation was supported by the primary care trust at that time (NHS Southampton City) and by the

Southampton City Council. The control areas were Gosport and Havant which had 14 SSCCs between them. These areas were selected because they had similar demographic features to those of Southampton.

The Healthy Conversation Skills (HCS) training intervention equipped staff with five core skills to help women address barriers to behaviour change and set goals for: reflection on current practice; asking 'open discovery' questions that usually began with 'what' or 'how' and encouraging the recipient to reflect on their issue of concern and identify their own solutions; and goal setting using Specific Measurable Action-oriented Realistic Timed Evaluated Reviewed (SMARTER) planning. The delivery of the intervention is summarised in the text of box 1. We trained 148 staff working within 14 SSCCs. They included play workers, family support and community development workers.

We conducted a baseline survey of women in Southampton and women in Gosport and Havant before the start of training (between January and July 2009) to assess baseline levels of the main outcomes, namely dietary quality and physical activity, and important intermediate outcomes in the relationship between the intervention and the main outcomes, namely general self-efficacy, specific self-efficacy for healthy eating and physical activity, and sense of control over life.

Diet was assessed using a 20-item food frequency questionnaire (FFQ) which was developed from a larger FFQ. Data from the FFQ, used in the SWS, were used to produce a standardised z-score (with mean 0 and SD 1.0).[25] This score has been named the 'prudent diet score'. The 20-item FFQ gives closely comparable scores to the full FFQ and can be used in settings where administration of a longer FFQ is not feasible. Level of physical activity was assessed using the General Practice Physical Activity Questionnaire (GPPAQ), which allows for categorisation of physical activity according to four levels of intensity—level 4 representing the highest intensity.[26] A questionnaire approach was deemed most appropriate given the study setting and participants, and the GPPAQ was selected following pilot work that demonstrated that it had better face validity in the participant population than the International Physical Activity Questionnaire and the Recent Physical Activity Questionnaire.[27 28] Intermediate outcomes were assessed using validated questionnaires as described previously.[24] General self-efficacy was derived from mothers' response to the General Self-efficacy Scale.[29] Specific self-efficacy for healthy eating and exercising were assessed with validated measures.[30 31] Women's sense of control was derived from their responses to a nine-item scale to measure women's perceived control over life.[32]

Women were interviewed at baseline within SSCCs by trained fieldworkers—any woman attending one of the participating Centres was eligible to participate. The fieldworkers were not involved in the delivery of the training intervention, nor did they have any involvement in the work of SSCCs other than to carry out the surveys at baseline and follow-up. We took this approach to recruitment because any woman with children aged 5 years and under is eligible to attend Sure Start and local data suggested that around 70% of these women were engaged with Sure Start in Southampton at the time. A year after the completion of training (between April and October 2011), the women seen at baseline were contacted by telephone using the contact details they had given at baseline. Those who could be contacted and who were willing to participate were interviewed over the telephone with a repeat questionnaire in order to assess dietary quality, physical activity level and other covariates. The same team of trained fieldworkers carried out the baseline and follow-up surveys.

Data were collected to assess the effectiveness of intervention implementation, changes in staff practice resulting from the training, exposure of women to trained staff and the context (both local and national) in which the intervention was introduced.[33]

We calculated that a sample size of 200 at baseline and follow-up in each group would give 80% power to detect a 0.275 SD difference in change in outcome between the intervention and control groups at a 5% significance level, allowing for a correlation of 0.75 between women's diet or physical activity levels at baseline and follow-up. Knowing that women in this age group are very mobile, we allowed for a retention rate of around 40% and aimed to interview 500 women in each group at baseline.

---

**Box 1 Healthy Conversation Skills training**

Communication is enhanced through practitioners developing the skill of asking open-ended, or open discovery, questions—those that generally begin with 'how' and 'what'. Such healthy conversations allow a patient or client to explore an issue, identify barriers, and generate solutions that can be reviewed with the practitioner at their next encounter. Training aimed to increase self-efficacy and sense of control of both practitioners and their clients.

The five core skills are:
1. To be able to identify and create opportunities to hold 'healthy conversations';
2. To use open discovery questions;
3. To reflect on practice;
4. To listen rather than provide information;
5. To support goal setting through SMARTER planning (specific, measurable, action-oriented, realistic, timed, evaluated, reviewed goals).

Healthy Conversation Skills training consisted of three 3 h group sessions over 3–5 weeks to allow time for practising and reflecting on skills, delivered by a team of researchers experienced in group work and behaviour change. This was followed by a period of on-going support, including a phone call from one of the trainers to find out how skills are being implemented in practice, and a 3 h follow-up workshop approximately three months after training. The phone call and workshop allow trainees to reflect on the training, how they implemented their new skills, any barriers to their implementation and plans for continued or increased use. They also allow the collection of data to evaluate the effectiveness of the training.

---

## Statistical analysis

Women's baseline characteristics between the intervention and control areas were compared using $\chi^2$ tests for categorical variables and $\chi^2$ tests for trend for ordered categorical variables. Continuous measures were checked for normality and then tested using t tests. Before and after comparisons were made in each site using matched-pairs t tests for continuous variables and Wilcoxon matched pairs signed-rank test for the Physical Activity Index which was in four categories. Multilevel modelling analysis was not feasible as the study comprised only two clusters—the intervention and control areas. In order to assess the difference in levels of outcome variables and covariates at follow-up, we compared data in the intervention and control groups using regression models and adjusting for the level of the corresponding variable at baseline.[34] We used linear regression for continuous outcomes, but for the Physical Activity Index, Poisson regression with robust variance was used to calculate the relative risk of being at the highest level (level 4) as opposed to any other level. Adjusting for baseline levels of variables also took account of the effects of the majority of factors that might confound the relationship of interest, including age and level of educational attainment. We have provided actual p values for the tests conducted, but, in the text, where we quote results as being non-significant, this relates to p values greater than 0.05.

## RESULTS

We surveyed 527 women in Southampton and 495 women in Gosport and Havant prior to the start of training. Similar numbers of women were followed up in the intervention and control areas giving data at both time points for 266 women from Southampton and 243 women from Gosport and Havant. The baseline characteristics of these women are shown in table 1.

The mean age in both groups was 32 years. Women in the intervention group had higher levels of educational attainment than women in the control group with 36% of Southampton women educated to degree level compared with 24.3% in Gosport and Havant. Although most women were white, a higher percentage of women were from other ethnic groups in Southampton (7.2%) than in Gosport and Havant (1.2%). Similar proportions of women were in receipt of benefits in the two areas, but women in the intervention group were more likely to own their homes and a higher proportion of women in the control group (85%) were registered with Sure Start than in the intervention group (63%).

The prudent diet scores of women in Southampton were significantly higher at baseline than those of the women in the control areas of Gosport and Havant—the mean prudent diet score was 0.2 (SD 1.0) for women in the intervention group compared with 0.0 (SD 0.9) for women in the control group. In contrast, levels of physical activity were higher in women in the control group

with 20.2% having a physical activity level of four compared with only 12.4% of women in the intervention group (p=0.01 for the trend across levels 1–4). Levels of self-efficacy (general efficacy and specific efficacy for healthy eating and physical activity) were similar in the two groups at baseline. Sense of control was higher in the Southampton women with a mean score of 27.6 (SD 2.8) compared with 27 (SD 2.5) in the Gosport and Havant women.

Table 2 compares levels of the main and intermediate outcomes (dietary quality, physical activity, self-efficacy and sense of control) at baseline and at follow-up in the intervention and control groups.

Dietary quality had declined between baseline and follow-up in both groups of women. The magnitude of the change was the same in both groups; both had a statistically significant 0.1 SD decline in dietary quality score and the adjusted difference (women's dietary quality score at follow-up taking account of baseline levels) was 0.0 (95% CI −0.11 to 0.12).

In a univariate analysis exploring predictors of the decline in dietary quality, lower sense of control and lower level of educational attainment were associated with a greater decline in dietary quality (β=0.11, 95% CI 0.07 to 0.15, p=0.00 and β=0.70, 95% CI 0.57 to 0.82, p=0.001, respectively). Lower levels of self-efficacy had a borderline significant relationship with decline in dietary quality (β=0.51, 95% CI −0.003 to 0.1069, p=0.06). In a multivariate regression analysis, the effects of sense of control and self-efficacy dropped out leaving educational attainment, which was highly correlated with sense of control and self-efficacy, as the strongest predictor of dietary decline (β=0.08, 95% CI 0.03 to 0.12, p=0.001), the relationship being driven by the women of lower educational attainment. Further analyses showed that the relationship between level of educational attainment and dietary quality decline was independent of food security and receipt of benefits. Using FFQ data, we assessed whether change in consumption of particular food groups was responsible for the decline in dietary quality. Patterns were complex, indicating increases in consumption of some unhealthy products such as pies, sausage rolls and crisps, but there were also some increases in healthy products including green salads. These findings applied to women in the intervention and control groups and there were few differences between the groups.

Self-efficacy and sense of control declined in both groups of women between baseline and follow-up and these changes, within each group, were statistically significant (p<0.001 and <0.0001, respectively, for both the intervention and control groups; table 2). The decline in self-efficacy and sense of control was smaller in the intervention group, indicating a benefit of the intervention. There were no significant differences, between the intervention and control groups, in the change in self-efficacy for eating healthy foods and for physical activity.

In order to establish whether the statistically significantly lower decline in sense of control and level of self-

**Table 1** Baseline characteristics of 243 women attending Gosport and Havant Sure Start Centres (control group) and 266 women attending Southampton Sure Start Centres (interventions group)

| Characteristics | Control (n=243) | Intervention (n=266) | p Value* |
|---|---|---|---|
| Age at baseline interview (yrs) (mean (SD)) | 32.0 (5.0) | 32.0 (5.8) | 0.9 |
| Educational attainment (n (%)) | | | 0.03 |
| None | 6 (2.5) | 11 (4.2) | |
| GCSE D or lower | 17 (7.0) | 12 (4.5) | |
| GCSE A*-C | 64 (26.3) | 58 (22.0) | |
| A-level | 85 (35.0) | 76 (28.8) | |
| HND | 12 (4.9) | 12 (4.5) | |
| Degree or above | 59 (24.3) | 95 (36.0) | |
| Number of children at baseline (n (%)) | | | 0.3 |
| 0 | 2 (0.8) | 2 (0.8) | |
| 1 | 116 (47.7) | 136 (51.1) | |
| 2 | 80 (32.9) | 88 (33.1) | |
| 3 | 29 (11.9) | 25 (9.4) | |
| 4+ | 16 (6.6) | 15 (5.6) | |
| Sure Start registered (n (%)) | | | <0.001 |
| No | 30 (12.3) | 62 (23.4) | |
| Yes | 206 (84.8) | 168 (63.4) | |
| Do not know | 7 (2.9) | 35 (13.2) | |
| In receipt of benefits (n (%)) | | | 0.5 |
| No | 162 (66.7) | 170 (63.9) | |
| Yes | 81 (33.3) | 96 (36.1) | |
| Home ownership (n (%)) | | | 0.001 |
| Owns or buying with mortgage | 164 (67.5) | 189 (71.1) | |
| Rents from private landlord | 35 (14.4) | 24 (9.0) | |
| Rents from council/housing association | 24 (9.9) | 46 (17.3) | |
| Other rented accommodation | 3 (1.2) | 1 (0.4) | |
| Lives with parents | 5 (2.1) | 5 (1.9) | |
| MOD/army property | 9 (3.7) | 0 (0.0) | |
| Other | 3 (1.2) | 1 (0.4) | |
| Ethnic group (n (%)) | | | 0.001 |
| White | 240 (98.8) | 246 (92.8) | |
| Non-white | 3 (1.2) | 19 (7.2) | |
| Food and money score (3 grps) (n (%)) | | | 0.03 |
| Food secure | 197 (81.4) | 234 (88.0) | |
| Food insecure | 24 (9.9) | 20 (7.5) | |
| Hungry | 21 (8.7) | 12 (4.5) | |
| General control: total (mean (SD)) | 27.0 (2.5) | 27.6 (2.8) | 0.02 |
| Paid work in past 7 days (n (%)) | | | 0.1 |
| No | 154 (63.4) | 185 (69.5) | |
| Yes | 89 (36.6) | 81 (30.5) | |
| Physical activity index (n (%)) | | | 0.01 |
| Level 1 | 93 (38.3) | 130 (48.9) | |
| Level 2 | 48 (19.8) | 41 (15.4) | |
| Level 3 | 53 (21.8) | 62 (23.3) | |
| Level 4 | 49 (20.2) | 33 (12.4) | |
| Prudent diet SD score (mean (SD)) | 0.0 (0.9) | 0.2 (1.0) | 0.04 |
| Self-efficacy (mean (SD)) | 14.9 (1.9) | 15.1 (1.7) | 0.2 |
| Efficacy in eating healthy foods (mean (SD)) | 14.5 (2.3) | 14.5 (2.3) | 0.96 |
| Efficacy in exercising (mean (SD)) | 12.3 (2.9) | 12.3 (2.8) | 0.95 |

*t Tests were used to assess differences in means. $\chi^2$ tests were used for categorical variables, and $\chi^2$ tests for trend for ordered categorical variables. Categories were merged before conducting $\chi^2$ tests where numbers were small.

efficacy in the intervention group could be attributed to a protective effect of the intervention, we explored the relationship of exposure (assessed by SSCC attendance in the 18 months before follow-up) with change in sense of control and self-efficacy. Higher levels of exposure were significantly associated with a smaller decline in

sense of control ($\beta=0.17$, 95% CI 0.05 to 0.29, p=0.006), but there was no association between exposure and level of self-efficacy.

The proportion of women in the intervention group who reported higher levels of activity (level 4) at follow-up was 22.9% compared with 12.4% at baseline

**Table 2** Comparison of outcome variables at baseline and follow-up and assessment of the difference at follow-up between the two groups adjusting for the baseline levels

| | Control | | | Intervention | | | Adjusted difference or relative risk (RR) | |
|---|---|---|---|---|---|---|---|---|
| | Baseline | Follow-up | p Value* | Baseline | Follow-up | p Value* | (95% CI)† | p Value |
| Physical activity index (n (%)) (RR) | | | 0.3 | | | 0.002 | 1.17 (0.86 to 1.60) | 0.3 |
| Level 1 | 93 (38.3) | 89 (36.6) | | 130 (48.9) | 105 (39.5) | | | |
| Level 2 | 48 (19.8) | 36 (14.8) | | 41 (15.4) | 49 (18.4) | | | |
| Level 3 | 53 (21.8) | 65 (26.8) | | 62 (23.3) | 51 (19.2) | | | |
| Level 4 | 49 (20.2) | 53 (21.8) | | 33 (12.4) | 61 (22.9) | | | |
| Prudent diet SD score (mean (SD)) | 0.0 (0.9) | −0.1 (0.9) | 0.052 | 0.2 (1.0) | 0.1 (1.0) | 0.005 | 0.00 (−0.11 to 0.12) | 0.9 |
| General self-efficacy (mean (SD)) | 14.9 (1.9) | 14.2 (1.5) | <0.001 | 15.1 (1.7) | 14.6 (1.5) | <0.001 | 0.26 (0.01 to 0.5) | 0.04 |
| Specific efficacy for healthy eating (mean (SD)) | 14.5 (2.3) | 14.4 (2.1) | 0.6 | 14.5 (2.3) | 14.2 (2.3) | 0.15 | −0.16 (−0,54 to 0.22) | 0.4 |
| Specific efficacy for exercising (mean (SD)) | 12.3 (2.9) | 12.5 (2.9) | 0.3 | 12.3 (2.8) | 12.4 (2.8) | 0.7 | −0.11 (−0.55 to 0.34) | 0.6 |
| Sense of control: total (mean (SD)) | 27.0 (2.5) | 25.5 (2.0) | <0.0001 | 27.6 (2.8) | 26.0 (2.0) | <0.0001 | 0.35 (0.05 to 0.65) | 0.02 |

*For the physical activity index, Wilcoxon matched pairs signed-rank test was used to test for differences between baseline and follow-up in each group. For all other variables a matched-pairs t test was used.
†This column gives the adjusted difference at follow-up between intervention and control sites, adjusted for baseline values.

(table 2). There was a smaller increase in physical activity among women in the control group, but this was not statistically significant (p>0.05). The adjusted difference between the groups, however, was not statistically significant. Change in physical activity was not predicted by sense of control, self-efficacy or levels of educational attainment.

We explored demographic changes between baseline and follow-up that might have accounted for the changes in the outcomes we had observed. The proportion of women with two or more children and the proportion in paid employment increased between baseline and follow-up (table 3). The change in the proportion of women with two or more children was of similar magnitude in the intervention and control groups with around 70% having two or more children at follow-up compared with around 50% at baseline. The proportion of women in paid employment increased more markedly in the intervention group than in the control group, although the increases were statistically significant within both groups as shown in table 3. Significantly more intervention group women than control group women

**Table 3** Numbers of children that women reported having and number of women in paid work at baseline and follow-up in each group

| | Control | | | Intervention | | |
|---|---|---|---|---|---|---|
| Baseline | Baseline | Follow-up | p Value* | Baseline | Follow-up | p Value* |
| Number of children (n (%)) | | | <0.0001 | | | <0.0001 |
| 0 | 2 (0.8) | 1 (0.4) | | 2 (0.8) | 3 (1.1) | |
| 1 | 116 (47.7) | 74 (30.5) | | 136 (51.1) | 75 (28.2) | |
| 2 | 80 (32.9) | 114 (46.9) | | 88 (33.1) | 139 (52.3) | |
| 3 | 29 (11.9) | 38 (15.6) | | 25 (9.4) | 35 (13.2) | |
| 4 | 14 (5.8) | 12 (4.9) | | 11 (4.1) | 10 (3.8) | |
| 5 | 2 (0.8) | 3 (1.2) | | 3 (1.1) | 2 (0.8) | |
| 6 | 0 (0.0) | 1 (0.4) | | 1 (0.4) | 1 (0.4) | |
| 7 | 0 (0.0) | 0 (0.0) | | 0 (0.0) | 1 (0.4) | |
| Paid work over the past 7 days (n (%)) | | | 0.0017 | | | <0.0001 |
| No | 154 (63.4) | 125 (51.4) | | 185 (69.6) | 122 (45.9) | |
| Yes | 89 (36.6) | 118 (48.6) | | 81 (30.5) | 144 (54.1) | |

*Wilcoxon matched pairs signed-rank test.

reported being in paid employment at follow-up: 31.2% compared with 23.5% (p=0.02).

The decline in dietary quality between baseline and follow-up was not explained by the increased number of children, increasing age of children or by being in paid work. In a fully adjusted regression model, the only significant predictor of dietary decline was educational attainment, a relationship described above. Being in paid work was, however, associated with higher self-efficacy at follow-up compared with baseline levels (β=0.31. 95% CI 0.055 to 0.57, p=0.017). Being in paid work was also strongly related to increased level of physical activity between baseline and follow-up (β=0.77, 95% CI 0.59 to 0.97, p<0.001) with women in employment having higher levels of activity.

## DISCUSSION
### Summary of findings
We observed significant reductions in the dietary quality of women in both the intervention and control groups between baseline and follow-up and in their levels of self-efficacy and sense of control, two factors known to predict dietary quality and physical activity. The decline in dietary quality was most marked in women of lower educational attainment, but was not explained by food insecurity or receipt of benefits. There were non-significant differences between the intervention and control groups in relation to the magnitude of the change in dietary quality. We found, however, that the magnitude of the decline in self-efficacy and sense of control was statistically significantly smaller in the intervention women from Southampton than in the women from the control areas of Gosport and Havant.

The difference in self-efficacy between the intervention and control groups was just under half a point (0.468) on a scale that ranged between 5 and 20, with an SD of 1.8, within the study population. The difference in sense of control was just under one point (0.875) on a scale that ranged between 9 and 36, with an SD of 2.5, within the study population. Previous research with women of childbearing age in Southampton has shown that a difference in sense of control of this size is associated with a significant increase in dietary quality.[35]

There was a statistically significant improvement in physical activity level between baseline and follow-up in the intervention group and a smaller increase in the control group that was not statistically significant. Improvement in the intervention group was not related to exposure to the intervention and the difference in the magnitude of change in physical activity level between the groups was not statistically significant. Improvement in physical activity was not predicted by women's level of educational attainment but was greatest among women who were in paid work at follow-up.

### Comparison with other research
A number of intervention studies have demonstrated that improved consumption of fruit and vegetables is mediated through improvements in people's self-efficacy and sense of control.[36 37] To the best of our knowledge, few studies have reported the evaluation of a complex intervention that has aimed to achieve behaviour change through training existing health and social care staff. While programmes like Women, Infants and Children (WIC) in the USA have used peer educators to bring about behaviour change, these educators were recruited to the intervention study specifically to deliver the intervention.[38] Other research groups have recognised the potential of training health professionals in order to bring about health behaviour change. A recent trial of disease prevention in primary care took a similar approach to the present study by training general practitioners and practice nurses in behaviour change counselling to use in their routine contacts with patients.[39] The aim of the study was to improve patients' health behaviours, including diet and physical activity. Consistent with our findings, the evaluation of the intervention revealed that it had positive effects on intermediate outcomes, including intention to change a behaviour and attempting to do so, but that there were no changes in the primary outcome of beneficial health behaviour change. The authors concluded that a single consultation with a trained clinician was unlikely to be sufficient to bring about behaviour change.[39]

### Strengths and limitations
The present study was innovative in delivering a training intervention to health and social care staff within existing services. Instead of delivering the intervention direct to the target population, this approach allowed staff who had received the training intervention to apply the behaviour change techniques they had learnt in their contacts with the target population—women of childbearing age from disadvantaged backgrounds. The nature of the intervention meant that it was sustainable and could continue to be delivered after the study was completed, particularly since it was delivered in partnership with local agencies. Indeed, the training programme has been commissioned by the local NHS community trust and the research team have trained health promotion staff within the trust in order that they can continue to deliver the training to new Sure Start staff and to update existing staff. Nevertheless, despite these benefits, evaluating a diffuse intervention of this type was challenging. One of the major challenges was the issue of exposure to the intervention. We had no control over women's attendance at SSCCs and it was not possible to ensure that participants surveyed at follow-up had been exposed to the intervention during the 1-year period since the training had been delivered. In this respect, this intervention study has some similarities with the challenges faced by natural experiments. A natural experiment evaluating a change in a health or community service would face similar issues to those faced in the present study where exposure to the

intervention cannot be controlled by the researchers, thus making exposure status difficult to determine.[40]

The design of the present complex public health intervention study was pragmatic. Randomisation at the level of SSCC was not feasible because of the frequent movement of staff between centres which would have led to contamination of control Centres. A cluster randomised controlled trial was considered but the need for many clusters, each being an entire local authority area, was considered unfeasible by the independent advisory group of experts. The control areas of Gosport and Havant were selected because they had similar demographic features to those of the intervention area—Southampton. In addition, the SSCCs in the intervention and control areas were sufficiently far apart geographically and managed by different local authorities, thus minimising the risk of contamination. All SSCCs in Southampton and in Gosport and Havant were included in the study and any women attending these centres were eligible to take part. Despite these attempts to select women with similar characteristics, there were some important differences between women in the intervention and control groups. While women in the two groups were similar in age, women in Southampton had higher levels of educational attainment than the women in the control group and, while similar proportions of women were in receipt of benefits, a higher proportion of women in Southampton owned their own home. The quality of diet was also better in the Southampton women than in the women from Gosport and Havant at baseline, although the reverse was true for physical activity: women in Gosport and Havant had higher physical activity levels than women in Southampton.

The measures used to assess the main and intermediate outcomes were all based on self-reported data. However, most of them had been shown to be valid measures within UK populations. The FFQ used to assess dietary quality was developed and validated with women of childbearing age within Southampton.[25] The measure of physical activity, the GPPAQ, has been recommended for use in UK populations by NICE and was selected for this study following pilot work which demonstrated its face validity with the study population.[26] The scale used to assess self-efficacy has been widely used throughout the world, including in the UK.[29] The scale for assessing sense of control has also been used around the world and,[32] in the UK, it had been used to assess sense of control in women participating in the Whitehall II study, although these women were aged 35–55 years of age and so older than participants in the present study.

Another strength of the present study was that participants were blind to their intervention status. This was achieved because women had no knowledge that they were part of an intervention study. Fieldworkers conducting the baseline and follow-up surveys were entirely separate from the staff delivering the intervention, although they could not be blinded to the intervention status of women.

## Interpretation of findings

The dietary quality of both groups of women declined in the 18-month period between the baseline and follow-up survey, with reductions greatest among women of lower educational attainment. The intermediate outcomes, sense of control and self-efficacy, also declined between baseline and follow-up, but the magnitude of their decline was significantly smaller in the intervention group than in the control group, suggesting a protective effect of the intervention. The fact that greater exposure to the intervention was associated with smaller decline in sense of control appears to confirm a protective effect of the intervention on women's sense of control, although the same trend was not seen for self-efficacy.

It is possible that worsening economic conditions might partly explain the decline in dietary quality, self-efficacy and sense of control among the study participants, particularly given the greater decline in these factors among women of lower educational attainment. Although receipt of benefits and food security was unrelated to the decline in dietary quality, self-efficacy or sense of control, these factors might be rather crude measures of women's economic circumstances We found that the proportion of women with two or more children increased significantly between baseline and follow-up in both groups and we know that any children in the baseline survey would have increased in age by at least 1 year by the time of the follow-up survey. While previous research on women of childbearing age has suggested that their self-efficacy declines as their children get older,[41] there were non-significant relationships between the number and age of children and self-efficacy and sense of control.

Our previous research has shown that self-efficacy and sense of control are predictive of dietary quality among women of childbearing age and that these psychosocial variables are also known to predict level of physical activity.[13] Self-efficacy and sense of control were intermediate factors in the relationship between the intervention and women's health behaviours within our conceptual model. The fact that women in the intervention group maintained a higher sense of control and self-efficacy than women in the control group is important and suggests that the intervention might have exerted a protective effect on psychological well-being.

Alongside the decline in dietary quality, we observed improvements in levels of physical activity among all women taking part in the study. There are a number of possible explanations for these contrasting patterns of health behaviour change. Significantly more women were in paid employment at follow-up than at baseline. Our measure of physical activity, the GPPAQ, records physical activity associated with work and our data indicate that physical activity levels were higher among women who reported being in paid work. Our findings might also be a reflection of the fact that the factors that facilitate changes in health behaviour might differ for diet and physical activity, not least because improving

diet is complex and involves eating less of some things like fat and sugar and more of others like fruit and vegetables, whereas improving physical activity involves increasing activity.

The right conditions were in place to bring about behaviour change: the majority of staff working within SSCCs in Southampton received training and the findings of the process evaluation conducted alongside the evaluation of outcomes have indicated that nearly all staff made significant improvements in their practice as a result of training. Despite this, we did not see positive changes in health behaviour among women attending Sure Start. There are a number of possible explanations for the lack of effectiveness of the intervention and the findings of the process evaluation shed light on these. Follow-up data on exposure have indicated that 43% of the women who took part in the follow-up survey were only making occasional attendances at Sure Start Centres during the period of follow-up, thus limiting their potential exposure to the staff who had been trained in the HCS intervention. Nevertheless, for the women who were attending Centres more regularly, exposure to the intervention appears to have had a protective effect on their sense of control. Women followed up were older and had higher levels of educational attainment than those who had been lost to follow-up, either because they declined to take part or because they were not contactable. The women followed up were also more likely to be in paid employment. These findings suggest that women who were more likely to benefit from use of Sure Start services because of their more disadvantaged circumstances were lost to follow-up. In most cases, loss to follow-up was because women could not be contacted on the phone numbers they had given during the baseline survey. It is possible that these women might have been living more chaotic lives and might have been a more mobile group than those followed up.

An intervention of this type, which is aiming to change the practice of a group of health and social care staff, is likely to take a number of years to embed in practice across an organisation. Added to this, it is likely that change in women's health behaviour will not come from a single 'healthy conversation' with a trained practitioner, but rather from repeated exposure to conversations over time.

## Implications for policy and practice

The intervention had a protective effect on sense of control, an important intermediate factor within our conceptual model for the relationship between exposure to HCS and change in diet and physical activity. This was achieved at a time when dietary quality, sense of control and self-efficacy declined among all study participants, perhaps due to economic circumstances. Conducting our research in Sure Start was challenging. It is the obvious setting in which to reach women from disadvantaged backgrounds. However, the difficulties of conducting research in this setting were considerable,

particularly given the mobility of the population served. Nevertheless, the improvements in staff practice resulting from the training intervention are an important finding and indicate that HCS training has the potential to improve health behaviour. That fact, combined with the protective effect of the intervention on sense of control among women in the intervention group, suggests that the intervention might have the potential to improve dietary quality and physical activity under more optimal circumstances. Thus, we believe that further studies to assess the efficacy and effectiveness of HCS are needed, but that these should be conducted in settings where contact with women is more certain, thus ensuring repeated exposure to the intervention as a component of antenatal care, for example.

Our findings demonstrate the difficulties and challenges of carrying out complex public health intervention research in a real-world setting. We attempted to assess the effects of the intervention under conditions that reflected the normal delivery of services—a true test of effectiveness. By doing so, however, it appears that we did not exert sufficient control over the implementation and evaluation of our intervention. The findings of our process evaluation show that our approach to recruitment might have led to only limited exposure to the intervention for some of the women that we followed up. The extent to which the implementation of complex interventions should be controlled has been widely debated and many have argued that it is not wise to over standardise an intervention since, although this might improve internal validity, it might also render the intervention ineffective.[42] In this case, our findings suggest that more control of the intervention and evaluation would have been justified to make the study work in a setting like Sure Start where attendance and participation are entirely voluntary. Alternative approaches to recruitment, for example, recruitment of women at the time of their enrolment onto a particular Sure Start activity and monitoring attendance at sessions would have been considerably more resource intensive and potentially difficult to fund.

Government policy to tackle obesity calls for action at community level to bring about improvements in health behaviour.[1] [2] Evidence of growing inequalities in health behaviour suggests the need for strategies that will target disadvantaged groups in community settings where they can be reached. Sure Start is one such setting. This study has demonstrated some of the challenges involved in implementing and evaluating complex interventions within a community setting like Sure Start. These challenges should not get in the way of community-level intervention research, but it is likely that more complex study designs will be required to overcome them. Policymakers and funders of research need to consider the impact of such challenges when deciding their research priorities.

**Author affiliations**
[1]MRC Lifecourse Epidemiology Unit, University of Southampton, Southampton, UK

[2]NIHR Nutrition Biomedical Research Centre, University of Southampton, Southampton, UK
[3]Public Health Team, NHS Southampton City, Civic Centre, Southampton, UK
[4]Primary Care and Population Sciences, Faculty of Medicine, University of Southampton, Southampton, UK

**Acknowledgements** The authors thank the staff of the Southampton City Council, NHS Southampton City and Hampshire County Council, in particular Sue Thompson, Liz Taylor, Stephanie Ramsay and Janet Hoff for their considerable help in facilitating the evaluation within the Sure Start Children's Centres. They are grateful to Jeanette Keyte for her contributions to the design and delivery of the training programme, to Vanessa Cox and the MRC LEU computing team for their invaluable help with the data, and to all working on the Southampton Initiative for Health.

**Contributors** All authors contributed to the conception and design of the study. MJ, JB, MB, TT, JD and RB were responsible for data collection. HI, JB, MB and CC conducted the data analysis. JB drafted the manuscript which was revised following comments from all authors. All authors have approved the final manuscript for publication. CC is guarantor.

**Funding** This study was funded by the Medical Research Council (UK), and the NIHR Nutrition Biomedical Research Centre, University of Southampton.

**Competing interests** None.

**Ethics approval** University of Southampton School of Medicine Ethics Committee.

**Provenance and peer review** Not commissioned; externally peer reviewed.

**Data sharing statement** No additional data are available.

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
