## [Reviewer comments · BMJ Open]

BMJ Open

What is the effect of a behaviour change intervention on the diets and physical activity levels of women attending Sure Start Children's Centres in Southampton? Findings from a non-randomised controlled trial

Journal:	BMJ Open
Manuscript ID:	bmjopen-2014-005290
Article Type:	Research
Date Submitted by the Author:	19-Mar-2014
Complete List of Authors:	Baird, Janis; University of Southampton, MRC Lifecourse Epidemiology Unit Jarman, Megan; University of Southampton, MRC Lifecourse Epidemiology Unit Lawrence, Wendy; University of Southampton, MRC Lifecourse Epidemiology Unit; University of Southampton, NIHR Nutrition Biomedical Research Centre Black, Christina; University of Southampton, MRC Lifecourse Epidemiology Unit Davies, Jenny; NHS Southampton City, Public Health Tinati, Tannaze; University of Southampton, MRC Lifecourse Epidemiology Unit Begum, Rufia; University of Southampton, MRC Lifecourse Epidemiology Unit; University of Southampton, NIHR Nutrition Biomedical Research Centre Mortimore, Andrew; NHS Southampton City, Public Health Robinson, S; University of Southampton, MRC Lifecourse Epidemiology Unit Margetts, Barrie; University of Southampton, Cooper, Cyrus; University of Southampton and Southampton University Hospitals NHS Trust, MRC Lifecourse Epidemiology Unit; University of Southampton, NIHR Nutrition Biomedical Research Centre Barker, Mary; University of Southampton, MRC Lifecourse Epidemiology Unit Inskip, Hazel; University of Southampton, MRC Lifecourse Epidemiology Unit
Primary Subject Heading:	Public health
Secondary Subject Heading:	Epidemiology
Keywords:	complex public health intervention, behaviour change, diet , physical activity, self efficacy, sense of control

For peer review only

**What is the effect of a behaviour change intervention on the diets and physical activity levels of**
**women attending Sure Start Children's Centres in Southampton? Findings from the evaluation of a**
**complex public health intervention**

Janis Baird,¹ Megan Jarman,^{1,2} Wendy Lawrence,^{1,2} Christina Black,¹ Jenny Davies,³ Tannaze
Tinati,¹ Rufia Begum,^{1,2} Andrew Mortimore,³ Sian Robinson,¹ Barrie Margetts,⁴ Cyrus Cooper,^{1,2} Mary
Barker,¹ Hazel Inskip.¹

¹ MRC Lifecourse Epidemiology Unit, University of Southampton, Southampton SO16 6YD

² NIHR Nutrition Biomedical Research Centre, University of Southampton, Southampton, SO16 6YD

³ Public Health Team, NHS Southampton City, Civic Centre, Southampton SO14 7LTS

⁴ Primary Care and Population Sciences, Faculty of Medicine, University of Southampton,
Southampton SO16 6YD

**Corresponding author**

Janis Baird

jb@mrc.soton.ac.uk

MRC Lifecourse Epidemiology Unit

University of Southampton

SO16 6YD

02380 777624

Abstract**Objectives**

A major component of the UK government response to the obesity epidemic is action in communities to improve people's health behaviour. This study evaluated effects of a community intervention on dietary quality and physical activity levels of women from disadvantaged backgrounds.

Design

Non-randomised controlled evaluation of a complex public health intervention.

Participants

527 women attending Sure Start Children's Centres (SSCC) in Southampton (intervention) and 495 women attending Centres in Gosport and Havant (control).

Intervention

Training SSCC staff in behaviour change skills that would empower women to change their health behaviours.

Outcomes

Main outcomes were dietary quality and physical activity. Intermediate outcomes were self-efficacy and sense of control.

Results

One year post-training, intervention staff used skills to support behaviour change significantly more than control staff. There were significant reductions of 0.1 SD in dietary quality of women in the intervention and control groups between baseline and follow-up and reductions in self-efficacy and sense of control. The decline in self-efficacy and control was significantly smaller in intervention than control group women (adjusted differences in self efficacy and control respectively 0.26 (95% CI 0.001, 0.50) and 0.35 (0.05, 0.65)). Lower decline in control was associated with higher levels of exposure in women in the intervention group. There was a statistically significant improvement in physical activity in the intervention group; 22.9% of women reporting highest level of physical activity compared with 12.4% at baseline, and smaller improvement in the control group. The difference in change in physical activity level between the groups was not statistically significant (adjusted difference 1.02 (0.74, 1.41)).

Conclusion

While the intervention did not improve women's diets and physical activity levels, it had a protective effect on intermediate factors - control and self efficacy, suggesting more prolonged exposure to the intervention may improve health behaviour and that further evaluation in a more controlled setting is justified.

Keywords

Complex public health intervention

Behaviour change

Diet

Physical activity

Self-efficacy

Sense of control

Women

[revised manuscript text omitted]

standardised score (with mean 0 and standard deviation 1.0).²⁵ This score has been named the
'prudent diet score'. The 20-item FFQ gives closely comparable scores to the full FFQ and can be
used in settings where administration of a longer FFQ is not feasible. Physical activity was assessed
using the General Practice Physical Activity Questionnaire (GPPAQ), which allows categorisation of
physical activity according to four levels of intensity – levels 4 representing the highest intensity.²⁶ A
questionnaire approach was deemed most appropriate given the study setting and participants, and
the GPPAQ was selected following pilot work that demonstrated that it had better face validity in
the participant population than the International Physical Activity Questionnaire and the Recent
Physical Activity Questionnaire.^{27,28} Intermediate outcomes were assessed using validated
questionnaires as described previously.²⁴ General self-efficacy was derived from mother's response
to the General Self-efficacy Scale.²⁹ Specific self-efficacy for healthy eating and exercising were
assessed with validated measures.^{30,31} Women's sense of control was derived from their responses
to a nine-item scale to measure women's perceived control over life.³²

[revised manuscript text omitted]

Figure 1 shows the mean dietary quality score of each group of women at baseline and follow up, the downward slope of each line indicating decline in dietary quality.

Figure 1: The dietary quality score of women in the intervention and control groups at baseline and follow up

Dotted line = Control group, continuous line = Intervention group

In a univariate analysis exploring predictors of the decline in dietary quality, lower sense of control, and lower level of educational attainment were associated with greater decline in dietary quality ($\beta=0.1126$, 95% CI 0.718, 0.1535, $p=0.000$ and $\beta=0.7009$, 95% CI 0.5770, 0.8247, $p=0.001$ respectively). Lower levels of self efficacy had a borderline significant association with decline in dietary quality ($\beta=0.517$, 95% CI -0.0034, 0.1069, $p=0.06$). In a multivariate regression analysis the effects of sense of control and self efficacy dropped out leaving educational attainment, which is highly correlated with sense of control and self efficacy, as the strongest predictor of dietary decline ($B=0.0784$, $p=0.001$), the association being driven by the women of lower educational attainment. Further analyses showed that the relationship between level of educational attainment with dietary quality decline was independent of food security and receipt of benefits. Using FFQ data we assessed whether change in consumption of particular food groups was responsible for the decline in dietary quality. Patterns were complex indicating increases in consumption of some unhealthy products such as pies, sausage rolls and crisps but also some increases in healthy products including green salads. These findings applied to women in the intervention and control groups and there were few differences between the groups.

Self efficacy and sense of control declined in both groups of women between baseline and follow up and these changes, within each group, were statistically significant (table 2, page 12).

Figures 2 and 3 show the same form of plot as in Figure 1 but the outcome measures displayed are for general self-efficacy and sense of control respectively. In both intervention and control groups these two measures declined but the decline was smaller in the intervention group, indicating a benefit of the intervention. There were no significant differences, between the intervention and control groups, in the change in self-efficacy for eating healthy foods and for physical activity.

Figure 2: The self efficacy score of women in the intervention and controls groups at baseline and follow up

Dotted line = Control group, continuous line = Intervention group

Figure 3: The sense of control score of women in the intervention and control groups at baseline and follow up

Dotted line = Control group, continuous line = Intervention group

In order to establish whether the significantly lower decline in sense of control and level of self efficacy in the intervention group could be attributed to a protective effect of the intervention, we explored the relationship of exposure (assessed by SSCC attendance in the 18 months prior to follow up) with change in sense of control and self efficacy. Higher levels of exposure were significantly associated with a smaller decline in sense of control ($\beta=0.1717$, 95% CI 0.0487, 0.2947, $p=0.006$) but there was no association between exposure and level of self efficacy. Further analysis revealed that the effect of exposure

Figure 4: Proportions of women at each level of Physical Activity Index according to time period and group

Figure 4 shows the physical activity level of each woman at follow up according to her physical activity level at baseline. The proportion of women in the intervention group who reported higher levels of activity (level 4) at follow up was 22.9% compared with 12.4% at baseline. There was a smaller increase in physical activity among women in the control group and this was not statistically significant ($p > 0.05$). Table 2 shows however, that the adjusted difference between the groups was not statistically significant. Change in physical activity was not predicted by sense of control, self efficacy or levels of educational attainment.

We explored demographic changes between baseline and follow up that may have accounted for the changes in the outcomes we had observed. The proportion of women with two or more children and the proportion in paid employment increased between baseline and follow up (table 3). The change in the proportion of women with two or more children was of similar magnitude in the intervention and control groups with around 70% having two or more children at follow up compared with around 50% at baseline. The proportion of women in paid employment increased

[revised manuscript text omitted]

this research.

**Conflict of interest**

The authors declare no competing interests.

The Corresponding Author has the right to grant on behalf of all authors and does grant on behalf of
all authors, an exclusive licence (or non exclusive for government employees) on a worldwide basis
to the BMJ Publishing Group Ltd to permit this article (if accepted) to be published in BMJ editions
and any other BMJPGJ products and sub-licences such use and exploit all subsidiary rights, as set out
in their licence.

12 13 **Author Contribution**

All authors contributed to the conception and design of the study. MJ, JB, MEB, TT, JD and RB were
responsible for data collection. HMI, JB, MEB and CC conducted the data analysis. JB drafted the
manuscript which was revised following comments from all authors. All authors have approved the
final manuscript for publication.

Cyrus Cooper is guarantor for the study.

24 **Ethical approval**

The research was given local ethical approval and was carried out in accordance with universal
ethical principles.

[revised manuscript text omitted]

- 39. Spanou C, Simpson SA, Hood K et al. Preventing disease through opportunistic, rapid
- engagment by primary care teams using behaviour change counselling (PRE-EMPT): protocol
- for a general-practice based cluster randomised trial. *BMC Family Practice* 2010;11:69.
- 40. Campbell K, Hesketh K, Silverii A et al. Maternal self efficacy regarding children's eating and
- sedentary behaviours in the early year: Associations with children's food intake and
- sedentary behaviour. *Internal Journal of Pediatric Obesity* 2010;5:501-508.

- 41. Craig P, Cooper C, Gunnell D et al. Using natural experiments to evaluate population health
- interventions: guidance for producers and users of evidence. Medical Research Council
- 2011.
- 42. Hawe P, Shiell A, Riley T. Complex interventions: how 'out of control' can a
- randomised controlled trial be? BMJ 2004;328:1561-3

For peer review only

BMJ Open

The effect of a behaviour change intervention on the diets and physical activity levels of women attending Sure Start Children's Centres: results from a complex public health intervention

Journal:	BMJ Open
Manuscript ID:	bmjopen-2014-005290.R1
Article Type:	Research
Date Submitted by the Author:	24-Jun-2014
Complete List of Authors:	Baird, Janis; University of Southampton, MRC Lifecourse Epidemiology Unit Jarman, Megan; University of Southampton, MRC Lifecourse Epidemiology Unit Lawrence, Wendy; University of Southampton, MRC Lifecourse Epidemiology Unit; University of Southampton, NIHR Nutrition Biomedical Research Centre Black, Christina; University of Southampton, MRC Lifecourse Epidemiology Unit Davies, Jenny; NHS Southampton City, Public Health Tinati, Tannaze; University of Southampton, MRC Lifecourse Epidemiology Unit Begum, Rufia; University of Southampton, MRc Lifecourse Epidemiology Unit; University of Southampton, NIHR Nutriton Biomedical Research Centre Mortimore, Andrew; NHS Southampton City, Public Health Robinson, S; University of Southampton, MRC Lifecourse Epidemiology Unit Margetts, Barrie; University of Southampton, Primary Care and Population Sciences, Faculty of Medicine Cooper, Cyrus; University of Southampton and Southampton University Hospitals NHS Trust, MRC Lifecourse Epidemiology Unit; University of Southampton, NIHR Nutriton Biomedical Research Centre Barker, Mary; University of Southampton, MRC Lifecourse Epidemiology Unit Inskip, Hazel; University of Southampton, MRC Lifecourse Epidemiology Unit
Primary Subject Heading:	Public health
Secondary Subject Heading:	Epidemiology
Keywords:	complex public health intervention, behaviour change, diet , physical activity, self efficacy, sense of control

For peer review only

**The effect of a behaviour change intervention on the diets and physical activity**
**levels of women attending Sure Start Children's Centres: results from a**
**complex public health intervention**

Janis Baird,¹ Megan Jarman,^{1,2} Wendy Lawrence,^{1,2} Christina Black,¹ Jenny Davies,³ Tannaze
Tinati,¹ Rufia Begum,^{1,2} Andrew Mortimore,³ Sian Robinson,¹ Barrie Margetts,⁴ Cyrus Cooper,^{1,2} Mary
Barker,¹ Hazel Inskip.¹

¹ MRC Lifecourse Epidemiology Unit, University of Southampton, Southampton SO16 6YD

² NIHR Nutrition Biomedical Research Centre, University of Southampton, Southampton, SO16 6YD

³ Public Health Team, NHS Southampton City, Civic Centre, Southampton SO14 7LTS

⁴ Primary Care and Population Sciences, Faculty of Medicine, University of Southampton,
Southampton SO16 6YD

**Corresponding author**

Janis Baird

jb@mrc.soton.ac.uk

MRC Lifecourse Epidemiology Unit

University of Southampton

SO16 6YD

02380 777624

Abstract**Objectives**

The UK government's response to the obesity epidemic calls for action in communities to improve people's health behaviour. This study evaluated the effects of a community intervention on dietary quality and levels of physical activity of women from disadvantaged backgrounds.

Design

Non-randomised controlled evaluation of a complex public health intervention.

Participants

527 women attending Sure Start Children's Centres (SSCC) in Southampton (intervention) and 495 women attending SSCCs in Gosport and Havant (control).

Intervention

Training SSCC staff in behaviour change skills that would empower women to change their health behaviours.

Outcomes

Main outcomes: dietary quality and physical activity. Intermediate outcomes: self-efficacy and sense of control.

Results

One year post-training, intervention staff used skills to support behaviour change significantly more than control staff. There were statistically significant reductions of 0.1 SD in the dietary quality of all women between baseline and follow-up and reductions in self-efficacy and sense of control. The decline in self-efficacy and control was significantly smaller in intervention than control group women (adjusted differences in self-efficacy and control respectively 0.26 (95% CI 0.001 to 0.50) and 0.35 (0.05 to 0.65)). Lower decline in control was associated with higher levels of exposure in women in the intervention group. There was a statistically significant improvement in physical activity in the intervention group; 22.9% of women reporting highest level of physical activity compared with 12.4% at baseline, and smaller improvement in the control group. The difference in change in physical activity level between the groups was not statistically significant (adjusted difference 1.02 (0.74 to 1.41)).

Conclusion

While the intervention did not improve women's diets and physical activity levels, it had a protective effect on intermediate factors - control and self-efficacy, suggesting more prolonged exposure to the intervention might improve health behaviour. Further evaluation in a more controlled setting is justified.

Keywords

Complex public health intervention

Behaviour change

Diet

Physical activity

Self-efficacy

Sense of control

Women

[revised manuscript text omitted]

Cyrus Cooper is guarantor for the study.

**Conflict of interest**

The authors declare no competing interests.

The Corresponding Author has the right to grant on behalf of all authors and does grant on behalf of
all authors, an exclusive licence (or non exclusive for government employees) on a worldwide basis
to the BMJ Publishing Group Ltd to permit this article (if accepted) to be published in BMJ editions
and any other BMJ PGL products and sub-licences such use and exploit all subsidiary rights, as set out
in their licence.

**Funding**

This study was funded by the Medical Research Council (UK), and the NIHR Nutrition Biomedical
Research Centre, University of Southampton. The funders have no vested interest in the findings of
this research.

**Data sharing**

No additional data are available.

Ethical approval

The research was given local ethical approval and was carried out in accordance with universal ethical principles.

For peer review only

[revised manuscript text omitted]

**Corresponding author**

Janis Baird

jb@mrc.soton.ac.uk

MRC Lifecourse Epidemiology Unit

University of Southampton

SO16 6YD

02380 777624

Abstract

Objectives

~~A major component of t~~he UK government's response to the obesity epidemic ~~calls for is~~ action in communities to improve people's health behaviour. This study evaluated ~~the~~ effects of a community intervention on dietary quality and ~~levels of~~ physical activity ~~levels~~ of women from disadvantaged backgrounds.

Design

Non-randomised controlled evaluation of a complex public health intervention.

Participants

527 women attending Sure Start Children's Centres (SSCC) in Southampton (intervention) and 495 women attending ~~SSCCs~~Centres in Gosport and Havant (control).

Intervention

Training SSCC staff in behaviour change skills that would empower women to change their health behaviours.

Outcomes

Main outcomes: ~~were~~ dietary quality ~~and~~ physical activity. Intermediate outcomes: ~~were~~ self-efficacy ~~and~~ sense of control.

Results

One year post-training, intervention staff used skills to support behaviour change significantly more than control staff. There were ~~statistically~~ significant reductions of 0.1 SD in ~~the~~ dietary quality of ~~all~~ women ~~in the intervention and control groups~~ between baseline and follow-up and reductions in self-efficacy and sense of control. The decline in self-efficacy and control was significantly smaller in intervention than control group women (adjusted differences in ~~self-efficacy~~self-efficacy and control respectively 0.26 (95% CI 0.001 ~~to~~ 0.50) and 0.35 (0.05 ~~to~~ 0.65)). Lower decline in control was associated with higher levels of exposure in women in the intervention group. There was a statistically significant improvement in physical activity in the intervention group; 22.9% of women reporting highest level of physical activity compared with 12.4% at baseline, and smaller improvement ~~in the control group~~. The difference in change in physical activity level between the groups was not statistically significant (adjusted difference 1.02 (0.74 ~~to~~ 1.41)).

Conclusion

While the intervention did not improve women's diets and physical activity levels, it had a protective effect on intermediate factors - control and self-efficacy, suggesting more prolonged exposure to the intervention might improve health behaviour, and that further evaluation in a more controlled setting is justified.

Keywords

Complex public health intervention

Behaviour change

Diet

Physical activity

Self-efficacy

Sense of control

Women

[revised manuscript text omitted]

~~Women's baseline characteristics were compared using t-tests for comparison of means, chi-squared~~
~~tests for categorical variables and chi-squared for trend for ordered categorical variables. Multi-level~~
~~modelling analysis was not feasible as the study comprised only two clusters—the intervention and~~
~~control areas. In order to assess the difference in levels of outcome variables and covariates at~~
~~follow-up, we compared data in the intervention and control groups using linear regression and~~
~~adjusting for level of the corresponding variable at baseline.³⁴ Adjusting for baseline levels of~~

Formatted: Font: Italic

~~variables also took account of the effects of the majority of factors that might confound the relationship of interest, including age, and level of educational attainment.~~

Results

We surveyed 527 women in Southampton and 495 women in Gosport and Havant prior to the start of training. Similar numbers of women were followed up in the intervention and control areas giving data at both time points for 266 women from Southampton and 243 women from Gosport and Havant. The baseline characteristics of these women are shown in table 1.

For peer review only

Table 1. Baseline characteristics of 243 women attending Gosport and Havant Sure Start Centres (control group) and 266 women attending Southampton Sure Start Centres (interventions group)

Characteristics	Control (n=243)	Intervention (n=266)	P-value*
Age at baseline interview (yrs) (mean(SD))	32.0 (5.0)	32.0 (5.8)	0.9
Educational attainment (n(%))			0.03
None	6 (2.5)	11 (4.2)	
GCSE D or lower	17 (7.0)	12 (4.5)	
GCSE A*-C	64 (26.3)	58 (22.0)	
A-level	85 (35.0)	76 (28.8)	
HND	12 (4.9)	12 (4.5)	
Degree or above	59 (24.3)	95 (36.0)	
Number of children at baseline (n(%))			0.3
0	2 (0.8)	2 (0.8)	
1	116 (47.7)	136 (51.1)	
2	80 (32.9)	88 (33.1)	
3	29 (11.9)	25 (9.4)	
4+	16 (6.6)	15 (5.6)	
SureStart registered (n(%))			<0.001
No	30 (12.3)	62 (23.4)	
Yes	206 (84.8)	168 (63.4)	
Don't know	7 (2.9)	35 (13.2)	
In receipt of benefits (n(%))			0.5
No	162 (66.7)	170 (63.9)	
Yes	81 (33.3)	96 (36.1)	
Home ownership (n(%))			0.001
Owns or buying with mortgage	164 (67.5)	189 (71.1)	
Rents from private landlord	35 (14.4)	24 (9.0)	
Rents from Council/Housing Association	24 (9.9)	46 (17.3)	
Other rented accommodation	3 (1.2)	1 (0.4)	
Lives with parents	5 (2.1)	5 (1.9)	
MOD/Army property	9 (3.7)	0 (0.0)	
Other	3 (1.2)	1 (0.4)	
Ethnic group (n(%))			0.001
White	240 (98.8)	246 (92.8)	
Non-white	3 (1.2)	19 (7.2)	
Food and money score (3 grps) (n(%))			0.03
Food secure	197 (81.4)	234 (88.0)	
Food insecure	24 (9.9)	20 (7.5)	
Hungry	21 (8.7)	12 (4.5)	
General control: total (mean(SD))	27.0 (2.5)	27.6 (2.8)	0.02
Paid work in past 7 days (n(%))			0.1
No	154 (63.4)	185 (69.5)	
Yes	89 (36.6)	81 (30.5)	

Characteristics	Control (n=243)	Intervention (n=266)	P-value*
Physical Activity Index (n(%)) (RR)			0.01
Level 1	93 (38.3)	130 (48.9)	
Level 2	48 (19.8)	41 (15.4)	
Level 3	53 (21.8)	62 (23.3)	
Level 4	49 (20.2)	33 (12.4)	
Prudent diet SD score (mean(SD))	0.0 (0.9)	0.2 (1.0)	0.04
Self-efficacy (mean(SD))	14.9 (1.9)	15.1 (1.7)	0.2
Efficacy in eating healthy foods (mean(SD))	14.5 (2.3)	14.5 (2.3)	0.96
Efficacy in exercising (mean(SD))	12.3 (2.9)	12.3 (2.8)	0.95

*t-tests were used to assess differences in means. Chi-squared tests were used for categorical variables, and chi-squared tests for trend for ordered categorical variables. Categories were merged before conducting chi-squared tests where numbers were small.

Mean age in both groups was 32 years. Women in the intervention group had higher levels of educational attainment than women in the control group with 36.0% of Southampton women educated to degree level compared with 24.3% in Gosport and Havant. Although most women were white, a higher percentage of women were from other ethnic groups in Southampton (7.2%) than in Gosport and Havant (1.2%). Similar proportions of women were in receipt of benefits in the two areas but women in the intervention group were more likely to own their homes and a higher proportion of women in the control group (85.0%) were registered with Sure Start than in the intervention group (63.0%).

The prudent diet scores of women in Southampton were significantly higher at baseline than those of the women in the control areas of Gosport and Havant – mean prudent diet score was 0.2 (standard deviation (SD) 1.0) in intervention women compared with 0.0 (SD 0.9) for women in the control group. In contrast, levels of physical activity were higher in women in the control group with 20.2% having a physical activity level of four compared with only 12.4% of women in the intervention group (P=0.01 for the trend across levels 1 to 4). Level of self-efficacy (general efficacy and specific efficacy for healthy eating and physical activity) were similar in the two groups at baseline. Sense of control was higher in the Southampton women with a mean score of 27.6 (SD 2.8) compared with 27.0 (SD 2.5) in Gosport and Havant.

Table 2 compares levels of the main and intermediate outcomes (dietary quality, physical activity, self-efficacy and sense of control) at baseline and follow-up in the intervention and control groups.

Table 2. Comparison of outcome variables at baseline and follow-up and assessment of the difference at follow-up between the two groups adjusting for the baseline levels

	Control		P-value [†]	Intervention		P-value [†]	Adjusted difference	P-value
	Baseline	Follow-up		or relative odds ratio (95%CI)*				
Physical Activity Index (n(%)) (RR)			0.3			0.002	1.1702 (0.8674 to 1.6041)	0.349
Level 1	93 (38.3)	89 (36.6)		130 (48.9)	105 (39.5)			
Level 2	48 (19.8)	36 (14.8)		41 (15.4)	49 (18.4)			
Level 3	53 (21.8)	65 (26.78)		62 (23.3)	51 (19.2)			
Level 4	49 (20.2)	53 (21.8)		33 (12.4)	61 (22.9)			
Prudent diet SD score (mean(SD))	0.0 (0.9)	-0.1 (0.9)	0.052	0.2 (1.0)	0.1 (1.0)	0.005	0.00 (-0.11 to 0.12)	0.9
General self-efficacy (mean(SD))	14.9 (1.9)	14.23 (1.5)	<0.001	15.1 (1.7)	14.65 (1.5)	<0.001	0.26 (0.01 to 0.5)	0.04
Specific efficacy for healthy eating (mean(SD))	14.5 (2.3)	14.4 (2.1)	0.6	14.5 (2.3)	14.2 (2.3)	0.15	-0.16 (-0.54 to 0.22)	0.4
Specific efficacy for exercising (mean(SD))	12.3 (2.9)	12.5 (2.9)	0.3	12.3 (2.8)	12.4 (2.87)	0.7	-0.11 (-0.55 to 0.34)	0.6
Sense of control: total (mean(SD))	27.0 (2.5)	25.5 (2.0)	<0.0001	27.6 (2.8)	26.0 (2.0)	<0.0001	0.35 (0.05 to 0.65)	0.02

*This column gives the adjusted difference at follow-up between intervention and control sites, adjusted for baseline values.

†For the Physical Activity Index, Wilcoxon matched pairs signed rank test was used to test for differences between baseline and follow-up in each group. For all other variables a matched-pairs t-test was used.

Dietary quality had declined between baseline and follow-up in both groups of women. The magnitude of the change was the same in both groups; both had a statistically significant 0.1 SD decline in dietary quality score and the adjusted difference (women’s dietary quality score at follow-up taking account of baseline levels) was 0.0 (95% CI -0.11 to 0.12).

Figure 1 shows the mean dietary quality score of each group of women at baseline and follow-up; the downward slope of each line indicating decline in dietary quality.

Figure 1: The dietary quality score of women in the intervention and control groups at baseline and follow up

Formatted: Font: Bold

Dotted line = Control group, continuous line = Intervention group

In a univariate analysis exploring predictors of the decline in dietary quality, lower sense of control, and lower level of educational attainment were associated with greater decline in dietary quality

($\beta=0.1126$, 95% CI 0.0718 to 0.1535, $P=0.000$ and $\beta=0.7009$, 95% CI 0.5770 to 0.8247,

$P=0.001$ respectively). Lower levels of self-efficacy had a borderline significant

association with decline in dietary quality ($\beta=0.517$, 95% CI -0.0034 to 0.1069, $P=0.06$).

In a multivariate regression analysis, the effects of sense of control and self-efficacy dropped out leaving educational attainment, which was highly correlated with sense of control and self-efficacy, as the strongest predictor of dietary decline ($\beta=0.08784$, 95% CI 0.03 to 0.12,

$P=0.001$), the relationship being driven by the women of lower educational attainment.

Further analyses showed that the relationship between level of educational attainment and dietary quality decline was independent of food security and receipt of benefits. Using FFQ data we assessed whether change in consumption of particular food groups was responsible for the decline in dietary quality. Patterns were complex indicating increases in consumption of some unhealthy products such as pies, sausage rolls and crisps but also some increases in healthy products including

Formatted: Font: Italic

Formatted: Font: Italic

Formatted: Font: Italic

Formatted: Font: Italic

Formatted: Font: Italic

Formatted: Font: Italic

Formatted: Font: Italic

Formatted: Font: Italic

green salads. These findings applied to women in the intervention and control groups and there were few differences between the groups.

Self-efficacy and sense of control declined in both groups of women between baseline and follow-up and these changes, within each group, were statistically significant ($P < 0.001$ and $P < 0.0001$ respectively for both intervention and control groups) (Table 2, page 12).

Figures 2 and 3 show the same form of plot as in Figure 1 but the outcome measures displayed are for general self-efficacy and sense of control respectively. In both intervention and control groups these two measures declined but the decline in self-efficacy and sense of control was smaller in the intervention group, indicating a benefit of the intervention. There were no significant differences, between the intervention and control groups, in the change in self-efficacy for eating healthy foods and for physical activity.

Figure 2: The self-efficacy score of women in the intervention and controls groups at baseline and follow-up

Formatted: Font: Bold

Dotted line = Control group, continuous line = Intervention group

Figure 3: The sense of control score of women in the intervention and control groups at baseline and follow up

Formatted: Font: Bold

Dotted line = Control group, continuous line = Intervention group

In order to establish whether the statistically significantly lower decline in sense of control and level of self-efficacy in the intervention group could be attributed to a protective effect of the intervention, we explored the relationship of exposure (assessed by SSCC attendance in the 18 months before prior to follow-up) with change in sense of control and self-efficacy. Higher levels of exposure were significantly associated with a smaller decline in sense of control ($\beta=0.1717$, 95% CI 0.05487 to 0.2947, $P=0.006$) but there was no association between exposure and level of self-efficacy. Further analysis revealed that the effect of exposure

Formatted: Font: Italic

Formatted: Font: Italic

Figure 4: Proportions of women at each level of Physical Activity Index according to time period and group

Formatted: Font: Bold

Figure 4 shows the physical activity level of each woman at follow-up according to her physical activity level at baseline. 
[revised manuscript text omitted]

Cyrus Cooper is guarantor for the study.

43 **Conflict of interest**

The authors declare no competing interests.

The Corresponding Author has the right to grant on behalf of all authors and does grant on behalf of
all authors, an exclusive licence (or non exclusive for government employees) on a worldwide basis
to the BMJ Publishing Group Ltd to permit this article (if accepted) to be published in BMJ editions
and any other BMJPG products and sub-licences such use and exploit all subsidiary rights, as set out
in their licence.

Funding

This study was funded by the Medical Research Council (UK), and the NIHR Nutrition Biomedical Research Centre, University of Southampton. The funders have no vested interest in the findings of this research.

Data Sharing

No additional data are available

Formatted: Font: Bold

Author Contribution

~~All authors contributed to the conception and design of the study. MJ, JB, MEB, TT, JD and RB were responsible for data collection. HMI, JB, MEB and CC conducted the data analysis. JB drafted the manuscript which was revised following comments from all authors. All authors have approved the final manuscript for publication.~~

~~Cyrus Cooper is guarantor for the study.~~

Ethical approval

The research was given local ethical approval and was carried out in accordance with universal ethical principles.

[revised manuscript text omitted]

Formatted: Left, Indent: Left: 0.5", Space Before: 0 pt, After: 10 pt, Line spacing: Multiple 1.15 li, No bullets or numbering, Tab stops: Not at 0" + 0.38"

39. ~~Spanou C, Simpson SA, Hood K et al. Preventing disease through opportunistic, rapid engagement by primary care teams using behaviour change counselling (PRE-EMPT): protocol for a general practice based cluster randomised trial. BMC Family Practice 2010;11:69.~~

40. Craig P, Cooper C, Gunnell D et al. Using natural experiments to evaluate population health interventions: guidance for producers and users of evidence. Medical Research Council 2011.

41. Campbell K, Hesketh K, Silverii A et al. Maternal self efficacy regarding children's eating and sedentary behaviours in the early year: Associations with children's food intake and sedentary behaviour. International Journal of Pediatric Obesity 2010;5:501-508.

~~42. Craig P, Cooper C, Gunnell D et al. Using natural experiments to evaluate population health interventions: guidance for producers and users of evidence. Medical Research Council 2011.~~

43-42. _____ Hawe P, Shiell A, Riley T. Complex interventions: how 'out of control' can a randomised controlled trial be? British Medical Journal 2004;328:1561-3

Formatted: Left, Indent: Left: 0.5", Space Before: 0 pt, After: 10 pt, Line spacing: Multiple 1.15 li, No bullets or numbering, Tab stops: Not at 0" + 0.38"

Formatted: Normal, Indent: Left: 0.25", No bullets or numbering